# Clustering earthquake signals and background noises in continuous seismic data with unsupervised deep learning

Léonard Seydoux [1 ✉], Randall Balestriero [2], Piero Poli[1], Maarten de Hoop[3], Michel Campillo[1] & Richard Baraniuk[2]

The continuously growing amount of seismic data collected worldwide is outpacing our abilities for analysis, since to date, such datasets have been analyzed in a human-expert-intensive, supervised fashion. Moreover, analyses that are conducted can be strongly biased by the standard models employed by seismologists. In response to both of these challenges, we develop a new unsupervised machine learning framework for detecting and clustering seismic signals in continuous seismic records. Our approach combines a deep scattering network and a Gaussian mixture model to cluster seismic signal segments and detect novel structures. To illustrate the power of the framework, we analyze seismic data acquired during the June 2017 Nuugaatsiaq, Greenland landslide. We demonstrate the blind detection and recovery of the repeating precursory seismicity that was recorded before the main landslide rupture, which suggests that our approach could lead to more informative forecasting of the seismic activity in seismogenic areas.

[1] ISTerre, équipe Ondes et Structures, Université Grenoble-Alpes, UMR CNRS 5375, 1381 Rue de la Piscine, 38610 Gières, France. [2] Electrical and Computational Engineering, Rice University, 6100 Main MS-134, Houston, TX 77005, USA. [3] Computational and Applied Mathematics, Rice University, 6100 Main MS-134, Houston, TX 77005, USA. ✉email: leonard.seydoux@univ-grenoble-alpes.fr

Current analysis tools for seismic data lack the capacity to investigate the massive volumes of data collected worldwide in a timely fashion, likely leaving crucial information undiscovered. The current reliance on human-expert analysis of seismic records is not only unscalable, but it can also impart a strong bias that favors the observation of already-known signals[1]. As a case in point, consider the detection and characterization of nonvolcanic tremors, which were first observed in the southwestern Japan subduction zone two decades ago[2]. The complex signals generated by such tremors are hard to detect in some regions due to their weak amplitude. Robustly detecting new classes of seismic signals in a model-free fashion would have a major impact in seismology (e.g., for the purpose of forecasting earthquakes), since we would better understand the physical processes of seismogenic zones (subduction, faults, etc.).

Recently, techniques from machine learning have opened up new avenues for rapidly exploring large seismic data sets with minimum a priori knowledge. Machine-learning algorithms are data-driven tools that approximate nonlinear relationships between observations and labels (supervised learning) or that reveal patterns from unlabeled data (unsupervised learning). Supervised algorithms rely on the quality of the predefined labels, often obtained via classical algorithms[3,4] or even manually[5–8]. Inherently, supervised strategies are used to detect or classify specific classes of already-known signals and, therefore, cannot be used for discovering new classes of seismic signals. Unsupervised tools are likely the best candidates to explore seismic data without using any explicit signal model, and hence discover new classes of seismic signals. For this reason, unsupervised methods are more relevant for seismology, where the data are mostly unlabeled and new classes of seismic signals should be sought. While supervised strategies are often easier to implement, thanks to the evaluation of a prediction error, unsupervised strategies mostly rely on implicit models that are challenging to design. Unsupervised learning-based studies have mostly been applied to the data from volcano-monitoring systems, where a large variety of seismo-volcanic signals are usually observed[9–12]. Some unsupervised methods have also been recently applied to induced seismicity[13,14], global seismicity[15], and local-vs-distance earthquakes[16]. In both cases (supervised or unsupervised), the keystone to success lies in the data representation, namely, we need to define an appropriate set of waveform features for solving the task of interest. The features can be manually defined[7,17,18] or learned with appropriates techniques such as artificial neural networks[3,5], the latter belonging to the field of deep learning.

In this paper, we develop a new unsupervised deep-learning method for clustering signals in continuous multichannel seismic time series. Our strategy combines a deep scattering network[19,20] for automatic feature extraction and a Gaussian mixture model for clustering. Deep scattering networks belong to the family of deep convolutional neural networks, where the convolutional filters are restricted to wavelets with modulus activations[19]. The restriction to wavelets filters allows the deep scattering networks to have explicit and physics-related properties (frequency band, timescales of interest, amplitudes) that greatly simplifies the architecture design in contrast with classical deep convolutional neural network. Scattering networks have shown to perform high-quality classification of audio signals[20–22] and electrocardiograms[23]. A deep scattering network decomposes the signal's structure through a tree of wavelet convolutions, modulus operations, and average pooling, providing a stable representation at multiple time and frequency scales[20]. The resulting representation is particularly suitable for discriminating complex seismic signals that may differ in nature (source and propagation effects) with several orders of different durations, amplitudes, and frequency contents. After decomposing the time series with the deep scattering network, we exploit the representation in a two-dimensional feature space that results from a dimension reduction for visualization and hence interpretation purposes. The two-dimensional features are finally fed to a Gaussian mixture model for clustering the different time segments.

The design of the wavelet filters have been conducted in many studies, and in each case led to data-adapted filterbanks based on intuition on the underlying physics[24–26] (e.g., music classification, speech processing, bioacoustics, etc.). In order to follow the idea of optimal wavelet design in a fully explorative way, we propose to learn the mother wavelet of each filterbank with respect to the clustering loss. By imposing a reconstruction constraint to the different layers of the deep scattering network, we guarantee to fully fit the data distribution together with improving the clustering quality. Our approach therefore preserves the structure of a deep scattering network while learning a representation relevant for clustering. It is an unsupervised representation learning method located in between the time-frequency analysis widely used in seismology and the deep convolutional neural networks. While classical convolutional networks usually require a large amount of the data for learning numerous coefficients, our strategy can still work with small data sets, thanks to the restriction to wavelet filters. In addition, the architecture of the deep scattering network is dictated by physical intuitions (frequency and timescales of interest). This is in contrast to the tedious task of designing deep convolutional neural networks, which today is typically pursued empirically.

In this study, we develop and apply our strategy to the continuous seismograms collected during the massive Nuugaatsiaq landslide[27]. We perform a short- and a long-term cluster analysis and identify many types of seismic signals. In particular, we identify long-duration storm-generated signals, accelerating percursory signals, and different other seismic events. Furthermore, we discuss key properties of our network architecture.

## Results

**Seismic records of the 2017 Nuugaatsiaq landslide**. We apply our strategy for clustering and detecting the low-amplitude precursory seismicity to the June 2017 landslide that occurred near Nuugaatsiaq, Greenland[28]. The volume of the rockfall was estimated between 35 and 51 million cubic meters by differential digital elevation models, forming a massive landslide[27]. This landslide triggered tsunami waves that impacted the small town of Nuugaatsiaq, and caused four injuries[27].

The continuous seismic wavefield was recorded by a three-component broadband seismic station (NUUG) located 30 km away from the landslide (Fig. 1a). We select the daylong three-component seismograms from June 17, 2017 00:00 to June 17, 2017 23:38 in order to disregard the mainshock signal (at 23:39) and focus on seismic data recorded before. A detailed inspection of the east component records revealed that a small event was occurring repetitively before the landslide, starting ~9 h before the rupture and accelerating over time[28,29]. The accelerating behavior of this seismicity suggests that an unstable initiation was at work before the landslide. This signal is not directly visible in raw seismic records; it is of weak amplitude, has a smooth envelope, and exhibits energy in between 2 and 8 Hz (Fig. 1b, c). While some of these events may be visible in the seismograms filtered between 2 and 8 Hz at times close to the landslide, a large part is hidden in the background noise. A proper identification of this signal cannot be done with classical detection routines such as STA/LTA (the ratio between the short-term and the long-term average of the seismogram[30]) because these techniques are only sensitive to sharp signal changes with decent signal-to-noise ratios[15], and do not provide information on waveform similarity.

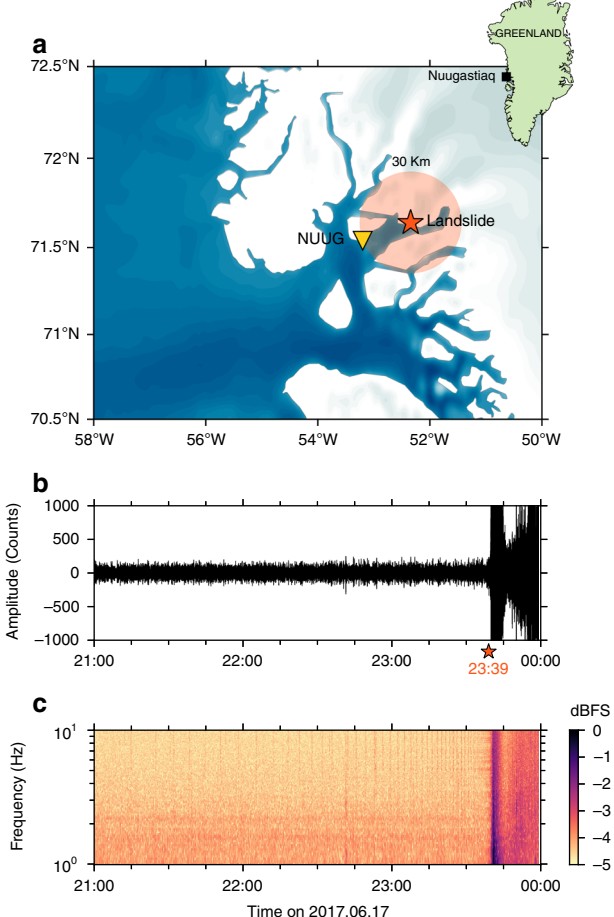

**Fig. 1 Geological context and seismic data. a** Location of the landslide (red star) and the seismic station NUUG (yellow triangle). The seismic station is located in the vicinity of the small town of Nuugaatsiaq, Greenland (top-right inset). **b** Raw record of the seismic wavefield collected between 21:00 UTC and 00:00 UTC on June 17, 2017. The seismic waves generated by the landslide main rupture are visible after 23:39 UTC. **c** Fourier spectrogram of the signal from **b** obtained over 35-s long windows.

These detection routines would potentially allow to detect a subset of these signals with many additional other signals, and would not allow to identify the accelerating behavior of these specific events. For this reason, the events were not investigated with STA/LTA, but with three-component template matching instead in ref. [28].

The template-matching strategy consists in a search for similar events in a time series with evaluating a similarity function (cross-correlation) between a predefined template event (often manually defined) and the continuous records. This method is sensitive to the analyzed frequency band, the template duration, and quality, making the template-matching strategy a severely supervised strategy, yet powerful[31]. Revealing this kind of seismicity with an unsupervised template-matching-based strategy could be done with performing the cross-correlation of all time segments (autocorrelation), testing every time segments as potential template event[32]. Considering that several durations, frequency bands, etc. should be tested, this approach is nearly impossible to perform onto large data sets for computational limitations[15].

In this study, we propose to highlight this precursory event in a blind way over a daylong, raw seismic record. Our goal is to show that even if the precursory signal was not visible after a detailed manual inspection of the seismograms, it could have been

correctly detected by our approach. The reader should bear in mind that clustering is an exploratory task[33]; we do not aim at overperforming techniques like template matching, but to provide the first preliminary statistical result that could simplify further detailed analyses, such as template selection for template-matching detection.

**Feature extraction from a learnable deep scattering network.** A diagram of the proposed clustering algorithm is shown in Fig. 2. The theoretical definitions are presented in "Methods". Our model first builds a deep scattering network that consists in a tree of wavelet convolutions and modulus operations (Eq. (5), "Methods"). At each layer, we define wavelet filterbank with constant quality factor from dilations and stretching of a mother wavelet (see Eq. (2), "Methods"). This is done according to a geometric progression in the time domain in order to cover a frequency range of interest. The input seismic signal is initially convolved with a first bank of wavelets, which modulus leads to a first-order scalogram (conv1), a time and frequency representation of one-dimensional signals widely used in seismology[34]. In order to speed up computations, we low-pass filter the coefficients in conv1, and perform a temporal downsampling (pool1) with an average-pooling operation[35]. The coefficients of pool1 are then convolved with a second wavelet bank, forming the second-order convolution layer (conv2). These succession of operations can be seen as a two-layer demodulation, where the input signal's envelope is extracted at the first layer (conv1) for several carrier frequencies, and where the frequency content of each envelope is decomposed again at the second layer (conv2)[20].

We define a deep scattering network as the sequence of convolution-modulus operations performed at higher orders, allowing to scatter the signal structure through the tree of time and frequency analyses. We finally obtain a locally invariant signal representation by applying an average-pooling operation to the all-order pooling layers[19–21]. This pooling operation is adapted for concatenation, with an equal number of time samples at each layer (Fig. 2). The scattering coefficients are invariant to local time translation, small signal deformations, and signal overlapping. They incorporate multiple timescales (at different layers) and frequencies scales (different wavelets). The tree of operations in a scattering network forms a deep convolutional neural network, with convolutional filters restricted to wavelets, and with modulus operator as activation function[19]. Scattering networks are located in between (1) classical time and frequency analysis routinely applied in seismology (2) deep convolutional neural networks where the unconstrained filters are often hard to interpret, and where the network architecture is often challenging to define. In contrast, deep scattering networks can be designed in a straightforward way, thanks to the analytic framework defined in ref. [19].

From one layer to another, we increase the filterbanks frequency range in order to consider at the same time small-duration details and larger-duration histories (see Table 1, case D for the selected architecture in this study). The number of wavelets per octaves and number of octaves define the frequency resolution and bandwidth of each layer. The scattering network depth (total number of layers) controls the final temporal resolution of the analysis. Following the recommendations cross-validated onto audio signal classification[20], we use a large number of filters at the first layer, and we gradually increase the number of octaves while reducing the number of wavelets per octave from the first to the last layer (Table 1, case D). That way, the representation is dense at the layer conv1 and gets sparser at the higher-order layers conv2 and conv3. This has the main effect of improving the contrast between signals of different nature[20].

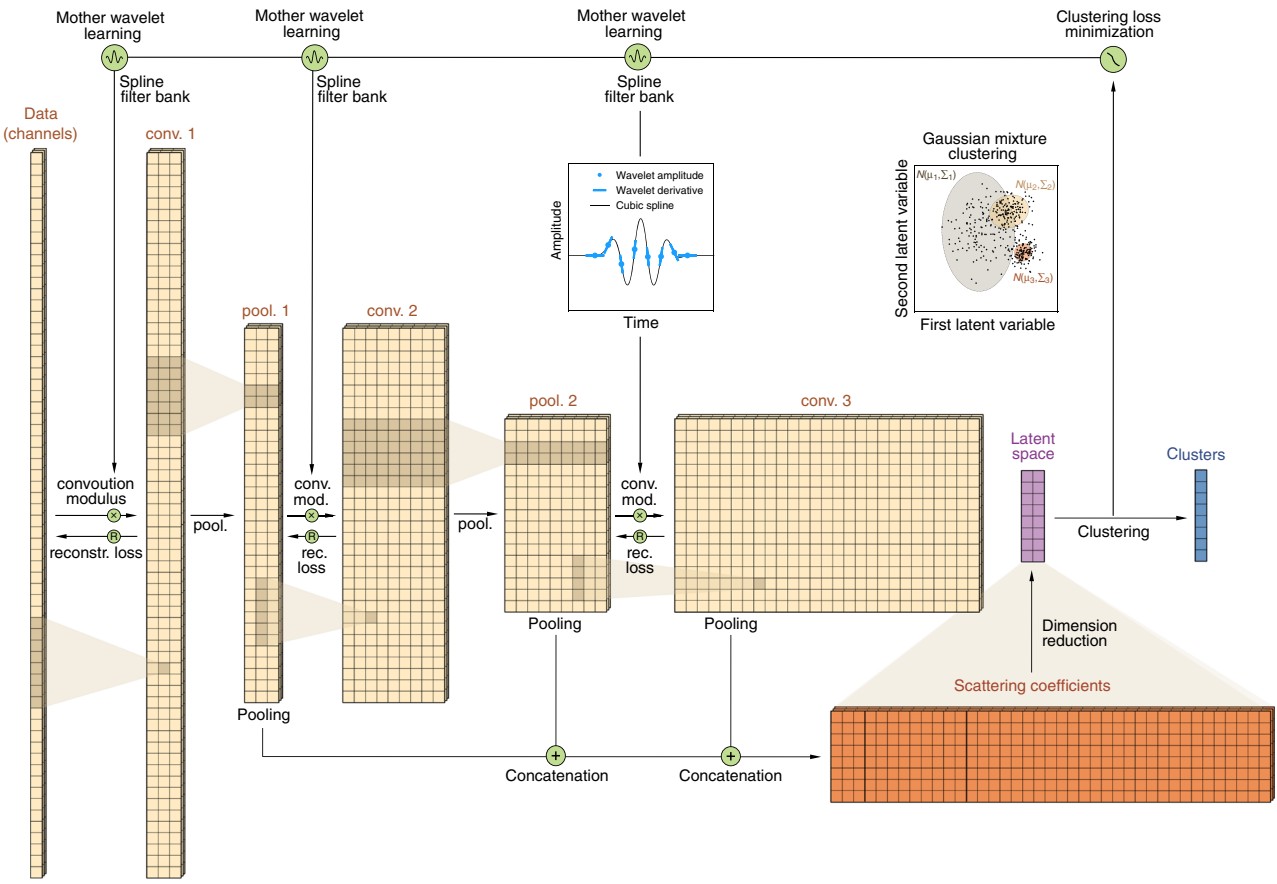

**Fig. 2 Deep learnable scattering network with Gaussian mixture model clustering.** The network consists in a tree of convolution and modulus operations successively applied to the multichannel time series (conv1--3). A reconstruction loss in calculated at each layer in order to constrain the network not to cancel out any part of the signal (Eq. (13), "Methods"). From one layer to another, the convolution layers are downsampled with an average-pooling operation (pool1--2), except for the last layer which can be directly used to compute the scattering coefficients. This allows to analyze large timescales of the signal structure with the increasing depth of the deep scattering network at a reasonable computational cost. The scattering coefficients are finally obtained from the equal pooling and concatenation of the pool layers, forming a stable high-dimensional and multiple time and frequency-scale representation of input multichannel time series. We finally apply a dimension reduction to the set of scattering coefficients obtained at each channel in order to form the low-dimensional latent space (here two-dimensional as defined in Eq. (10), "Methods"). We use a Gaussian mixture model in order to cluster the data in the latent space (Eq. (11), "Methods"). The negative log-likelihood of the clustering is used to optimize the mother wavelet at each layer (inset) with Adam stochastic gradient descent[39] described in Eq. (14) ("Methods"). The filterbank of each layer $\ell$ is then obtained by interpolating the mother wavelet in the temporal domain $\psi_0^{(\ell)}(t)$ with Hermite cubic splines (Eq. (9), "Methods"), and dilating it over the total number of filters $J^{(\ell)}Q^{(\ell)}$ (Eq. (2), "Methods").

**Table 1 Set of different tested parameters (with corresponding cumulative detection curves shown in Supplementary Fig. 1).**

| Ref. | Data | | Scattering network | | | | Learning | | | |
|---|---|---|---|---|---|---|---|---|---|---|
| | Start | End | $J^{(\ell)}$ | $Q^{(\ell)}$ | $\mathcal{K}$ | Pool. | Clusters | | Loss (clus.) | Loss (rec.) |
| A | 15:00 | 23:30 | 3, 6, 6 | 8, 2, 1 | 7 | $2^{10}$ | 10 → 4 | | 3.79 | 4.20 |
| B | 15:00 | 23:30 | 3, 6, 6 | 8, 2, 1 | 11 | $2^{10}$ | 10 → 3 | | 3.42 | 5.40 |
| C | 15:00 | 23:30 | 3, 6, 6 | 8, 2, 1 | 15 | $2^{10}$ | 10 → 3 | | 3.17 | 5.49 |
| ★D | **00:30** | **23:30** | **4, 6, 6** | **8, 4, 3** | **11** | **$2^{10}$** | **10 → 4** | | **2.96** | **3.06** |
| E | 00:30 | 23:30 | 3, 6, 6 | 8, 2, 1 | 11 | $2^{9}$ | 10 → 6 | | 3.67 | 1.76 |
| F | 00:30 | 23:30 | 3, 6, 6 | 8, 2, 1 | 11 | $2^{11}$ | 10 → 4 | | 3.11 | 3.06 |

The results presented in Figs. 3 and 4 are obtained with the set of parameters D (black star and bold typeface), with the lowest clustering loss. See the Supplementary Note 3 and Supplementary Fig. 1 for further details.

We finally choose the network depth based on the range of timescales of interest. In this study, we aim at investigating mostly impulsive earthquake-like signals that may last between several seconds to less than 1 min. A deeper scattering network could be of interest in order to analyze the properties of longer-duration signals, such as seismic tremors[36] or background seismic noise.

Finally, with our choice of pooling factors, we obtain a temporal resolution of 35 s for each scattering coefficient.

**Clustering seismic signals.** The scattering coefficients are built in order to be linearly separable[23] so that the need for a

high-dimensional scattering representation is greatly reduced. In fact, it is even possible to enforce the learning to favor wavelets that not only solve the task but also provide a lower-dimensional representation of the signal. We do so by reducing the dimension of the scattering coefficients with projection onto the first two principal components (Eq. (10), "Methods"). This also improves the data representation in two dimensions and eases the interpretation. More flexibility could be also obtained by using the latent representation of an autoencoder because autoencoders can lower the dimension of any data sets with nonlinear projections. However, such dimension reduction must be thoroughly investigated because it adds a higher-level complexity to the overall procedure (autoencoder learning rate, architecture, etc.), and will define the goal of future studies.

The two-dimensional scattering coefficients are used to cluster the seismic data. We use a Gaussian mixture model[37] for clustering, where the idea is to find the set of $K$-normal distributions of mean $\mu_k$ and covariance $\Sigma_k$ (where $k = 1\ldots K$) that best describe the overall data (Fig. 2 inset and Eq. (11), "Methods"). A categorical variable is also inferred in order to allocate each data sample into each cluster, which is the final result of our algorithm. Gaussian mixture model clustering can be seen as a probabilistic and more flexible version of the $K$-means clustering algorithm, where each covariance can be anisotropic, the clusters can be unbalanced in term of internal variance, and where the decision boundary is soft[37].

Initialized with Gabor wavelets[38], we learn the parameters governing the shape of the wavelets with respect to the clustering loss (Eqs. (7) and (8), "Methods") with the Adam stochastic gradient descent[39] (Eq. (14), "Methods"). The clustering loss is defined as the negative log-likelihood of the data to be fully described by the set of normal distributions. We define the wavelets onto specific knots, and interpolate them with Hermite cubic splines onto the same time basis of the seismic data for applying the convolution (see "Methods" for more details). We ensure that the mother wavelet at each layer satisfies the mathematical definition of a wavelet filter in order to keep all the properties of a deep scattering network[23]. We finally add a constraint on the network in order to prevent the learning to dropout some signals that make the clustering task hard (e.g., outlier signals). This is done by imposing a reconstruction loss from one layer to its parent signal, noticing that a signal should be reconstructed from the sum of the convolutions of itself with a wavelet filterbank (Eq. (13), "Methods").

The number of clusters is also inferred by our procedure. We initialize the Gaussian mixture clustering algorithm with a (relatively large) number $K = 10$ clusters at the first epoch, and let all of these components be used by the expectation–minimization strategy[37]. This is shown at the first epoch in the latent space in Fig. 3a, where the Gaussian component mean and covariance are shown in color with the corresponding population cardinality on the right inset. As the learning evolves, we expect the representation to change the coordinates of the two-dimensional scattering coefficients in the latent space (black dots), leading to Gaussian components that do not contribute anymore to fit the data distribution, and therefore to be automatically disregarded in the next iteration. We can therefore infer a number of clusters from a maximal value. At the first epoch (Fig. 3a), we observe that the seismic data samples are scattered in the latent space, and that the Gaussian mixture model used all of the ten components.

The clustering loss decreases with the learning epochs (Fig. 3c). We declare the clustering to be optimal when the loss stagnates (reached after ~7000 epochs). The learning is done with batch processing, a technique that allows for faster computation by randomly selecting smaller subsets of the data set. This also avoids falling into local minima (as observed ~3500 epochs), and guarantees to reach a stable minimum that does not evolve anymore after epoch 7000 (Fig. 3c). After 10,000 training epochs, as expected, the scattering coefficients have been concentrated around the clusters centroids (Fig. 3b). The set of useful components have been reduced to four, a consequence of a better learned representation due to the learned wavelets at the last epoch (Fig. 3d). The cluster colors range from colder to warmer colors, depending on the population size.

The clustering loss improves by a factor of ~4.5 between the first and the last epoch (Fig. 3c). At the same time, the reconstruction loss is more than 15 times smaller than at the first training epoch (Table 1). This indicates that the basis of wavelets filterbanks used in the deep scattering network is powerful to accurately represent the seismic data with ensuring a good-quality clustering at the same time.

**Analysis of clusters**. The temporal evolution of each clusters is presented in Fig. 4. The within-cluster cumulative detections obtained after training are presented in Fig. 4a for clusters 1 and 2, and in Fig. 4b for clusters 2 and 3. The two most populated clusters (1 and 2, Fig. 4a) gather >90% of the overall data (Fig. 3b). They both show a linear detection rate over the day with no particular concentration in time and, therefore, relate to the background seismic noise. Clusters 3 and 4 (Fig. 4b) show different nonlinear trends that include 10% of the remaining data.

The temporal evolution of cluster 4 is presented in Fig. 4b. The time segments that belong to cluster 4 are extracted and aligned to a reference event (at the top) with local cross-correlation for better readability (see Supplementary Note 1). These waveforms contain seismic events localized in time with relatively high signal-to-noise ratio and sharp envelope. These events do not show a strong similarity in time, but they strongly differ from the event belonging to other clusters, explaining why they have been gathered in the same cluster. The detection rate is sparse in time, indicating that cluster 4 is mostly related to a random background seismicity or other signals, interest in which is beyond the scope of this paper.

The temporal evolution of cluster 3 shows three behaviors. First, we observe a nearly constant detection rate from the beginning of the day to ~07:00. Second, the detection rate lowers between 07:00 and 13:00, where only 4% of the within-cluster detections are observed. An accelerating seismicity is finally observed from 13:00 up to the landslide time (23:39 UTC). The time segments belonging to cluster 3 are reported on Fig. 4d in gray colorscale, and aligned with local cross-correlation with a reference (top) time segment. The correlation coefficients obtained for the best-matching lagtime are indicated in orange color in Fig. 4e. As with the template-matching strategy, we clearly observe the increasing correlation coefficient with the increasing event index[28], indicating that the signal-to-noise ratio increases toward the landslide. This suggests that the repeating event may exist earlier in the data before 15:00, but that the detection threshold of the template-matching method is limited by the signal-to-noise ratio[28]. Because our clustering approach is probabilistic, it is possible that some time segments share sufficient similarity with the precursory events to have been placed in the same cluster. The pertinence of our approach could be further tested by similarity tests specific to the precursory signals, which is beyond the scope of this study. We note that the probability of these 171 events to belong to the same cluster remains high according to our clustering (Fig. 4e). We also note that 97% of the precursory events previously found[28] are recovered.

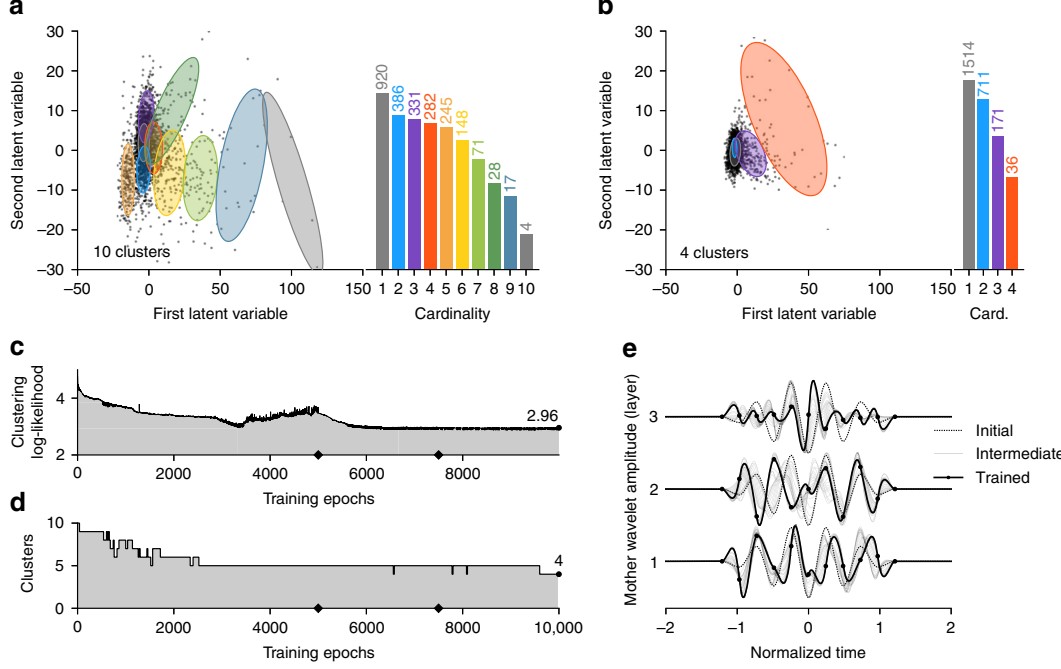

**Fig. 3 Learning results.** Scattering coefficients in the latent space at initialization (**a**) and after learning (**b**). The covariance of each component of the Gaussian mixture model is represented by a colored ellipse centered at each component mean. All of the ten components are used at the initial stage with a steadily decaying number of elements per clusters, while only four are used at the final stage with unbalanced population size. The clustering-negative log-likelihood (**c**, top) decreases with the learning epochs, indicating that the clustering quality is improved by the learned representation. We also observe that the reconstruction loss fluctuates and remains as low as possible (**c**, bottom). The number of cluster with respect to the increasing training epoch is shown in (**d**). Finally, the initial, intermediate, and final wavelets at each layer (**e**) are shown in the time domain interpolated from 11 knots.

An interesting observation is the change of behavior in the detection rate of this cluster at nearly 07:00 (Fig. 4b). The events that happened before 07:00 have all a relatively high probability to belong to cluster 3, refuting the hypothesis that noise samples have randomly been misclassified by our strategy (Fig. 4e). The temporal similarity of all these events in Fig. 4d is particularly visible for later events (high index) because the signal-to-noise ratio of these events increases toward the landslide[28]. The two trends may be either related to similar signals generated at same position (same propagation) with a different source, or by two types of alike-looking events that differ in nature, but that may have been gathered in the same cluster because they strongly differ from the other clusters. This last hypothesis can be tested with using hierarchical clustering[40]. Our clustering procedure highlighted those 171 similar events in a totally unsupervised way, without the need of defining any template from the seismic data. The stack of the 171 waveforms is shown in black solid line in Fig. 4d, indicating that the template of these events is defined in a blind way thanks to our procedure. In addition, these events have very similar properties (duration, seismic phases, envelope) in comparison with the template defined in ref. [28].

## Discussion

We have developed a novel strategy for clustering and detecting seismic events in continuous seismic data. Our approach extends a deterministic deep scattering network by learning the wavelet filterbanks and applying a Gaussian mixture model. While scattering networks correspond to a special deep convolutional neural network with fixed wavelet filterbanks, we allow it to fit the data distribution by learnability of the different mother wavelets; yet we preserve the structure of the deep scattering network allowing interpretability and theoretical guarantees. We combine the powerful representation of the learnable scattering network with Gaussian mixture clustering by learning the wavelet filters

according to the clustering loss. This allows to learn a representation of multichannel seismic signals that maximizes the quality of clustering, leading to an unsupervised way of exploring possibly large data sets. We also impose a reconstruction loss as each layer of the deep scattering network, following the ideas of convolutional autoencoders, and preventing to learn trivial solutions such as zero-valued filters.

Our strategy is capable of blindly recovering the small-amplitude precursory signal reported in refs. [28,29]. This indicates that waveform templates can be recovered from our method without the need of any manual inspection of the seismic data prior to the clustering process, and tedious selection of the waveform template in order to perform high-quality detection. Such unsupervised strategy is of strong interest for seismic data exploration, where the structure of seismic signals can be complex (low-frequency earthquakes, nonvolcanic tremors, distant vs. local earthquakes, etc.), and where some class of unknown signals is likely to be disregarded by a human expert.

In the proposed workflow, only a few parameters need be chosen, namely the number of octaves and wavelets per octave at each layer $J^{(\ell)}$ and $Q^{(\ell)}$, the number of knots $\mathcal{K}$ the pooling factors and the network depth $M$. This choice of parameters is extremely constrained by the underlying physics. The number of octaves at each layer controls the lowest analyzed frequency at each layer, and therefore, the largest timescale. The pooling factor and number of layers $M$ should be chosen according to the analyzed timescale at each layer, and the final maximal timescale of interest for the user. We discuss our choice of parameters with testing several parameter sets summarized in Table 1 and with the corresponding results presented in Supplementary Fig. 1 for the cumulative detection curves, within-cluster population sizes and learned mother wavelets (Supplementary Note 2). All the results obtained with different parameters show extremely similar cluster shapes in the time domain, and the precursory signal accelerating

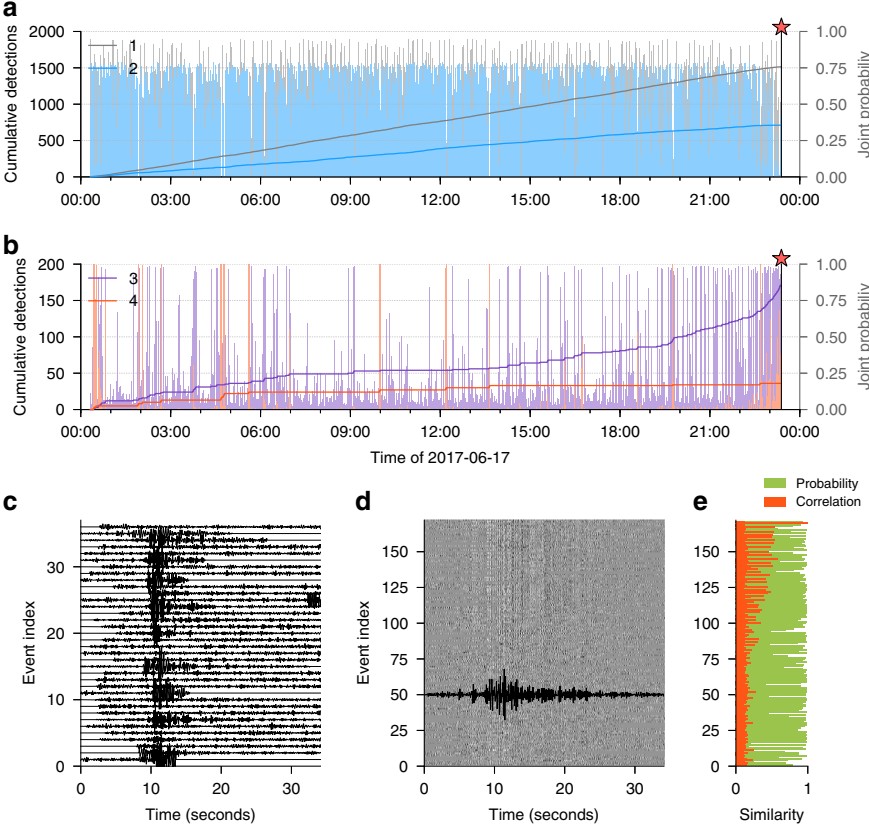

**Fig. 4 Analysis of clusters in the time domain.** Within-cluster cumulative number of detection of events in clusters 1 and 2 (**a**) and clusters 3 and 4 (**b**) at epoch 10,000. The relative probability for each time window to belong to each cluster is represented with lighter bars. The waveforms extracted within the last two clusters (purple and red) are extracted and aligned with respect to a reference waveform within the cluster, for cluster 4 (**c**) and cluster 3 (**d**). The seismic data have been bandpass-filtered between 2 and 8 Hz for better visualization of the different seismic events. **e** Similarity measurement in the time domain (correlation) and in the latent space (probability) for the precursory signal.

shape is always recovered. We see that a low number of 3 or 4 clusters are found in almost all cases, with a similar detection rates over the day. Furthermore, we observe that the shapes of the learned wavelets is stable for different data-driven tests, and in particular, the third-order wavelet is similar with all the tested parameters (Fig. 5g). This result makes sense because the coefficients that output from the last convolutional layer conv3 are overrepresented in comparison with the other ones. We also observe that the procedure still works with only a few amount of data (Fig. 5a–c), a very strong advantage compared with classical deep convolutional neural networks that often require a large amount of the data to be successfully applied.

Besides being adapted to small amount of the data, our strategy can also work with large data sets, as scalability is guaranteed by batch processing, and using only small-complexity operators (convolution and pooling). Indeed, batch processing allows to control the amount of data seen by the scattering network and GMM at a single iteration, each epoch being defined when the whole data set have been analyzed. There is no limitation to the total amount of the data being analyzed because only the selected segments at each iteration are fed to the network. At longer timescales, the number of clusters needed to fit the seismic data must change, however, with an expectation that the imbalance between clusters should increase. We illustrate this point with another experiment performed on the continuous seismogram recorded at the same station over 17 days, including the date of the landslide (from June 1, 2017 to June 18, 2017). With this larger amount of the data, the clustering procedure still converges and exhibit nine new clusters. The hourly within-clusters

detections of these new clusters are presented in Fig. 5. Among the different clusters found by our strategy, we observe that >93% of the data are identified in slowly evolving clusters, most likely related to fluctuations of the ambient seismic noise (Fig. 5, clusters A to E). The most populated clusters (A and B) occupy >61% of the time, and are most likely related to a diffuse wavefield without any particular dominating source. Interestingly, we observe two other clusters with large population with a strong localization in time (clusters C and D in Fig. 5). A detailed analysis of the ocean-radiated microseismic energy[41,42] allowed us to identify the location and dominating frequency of the sources responsible for these clusters to be identified (explained in Supplementary Note 3 and illustrated in Supplementary Figs. 2 and 3). The seismic excitation history provided by these oceanographic models of the best-matching microseismic sources have been reported on clusters C and D in Fig. 5.

Compared with these long-duration clusters, the clustering procedure also reports very sparse clusters where <7% of the seismic data are present. Because of clustering instabilities caused by the large class imbalance of the seismic data, we decided to perform a second-order clustering on the low-populated clusters. This strategy follows the idea of hierarchical clustering[40], where the initially identified clusters are analyzed several consecutive times in order to discover within-cluster families. For the sake of brevity, we do not intend to perform a deep-hierarchical clustering in this paper, but to illustrate the potential strength of such strategy in seismology, where the data are essentially class-imbalanced. We perform a new clustering from the data obtained in the merged low-populated clusters (F to I in Fig. 5). This

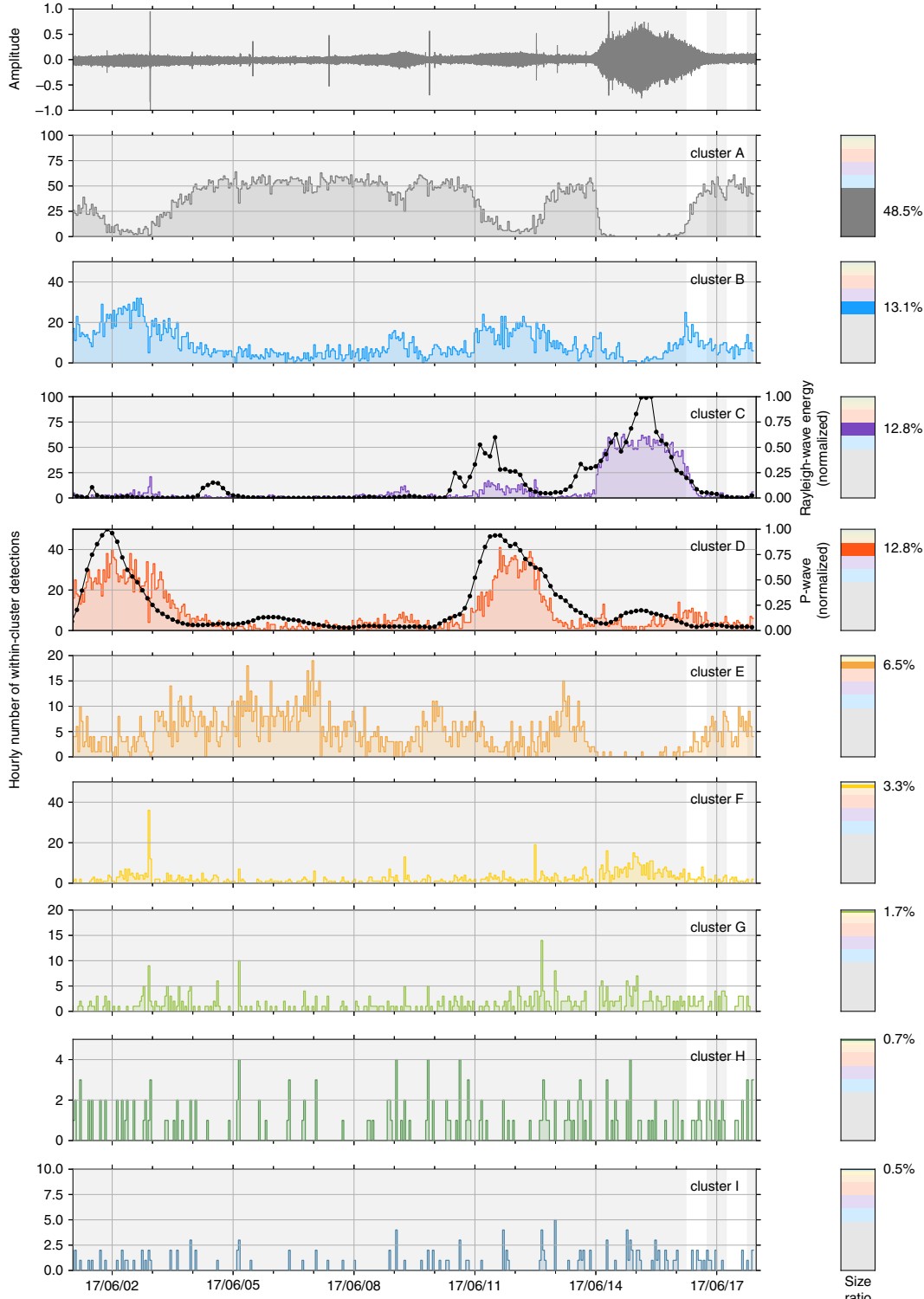

**Fig. 5 Clustering results obtained from long-duration seismic data.** The broadband seismogram recorded by the station NUUG (Fig. 1) is presented in the top plot. The hourly within-cluster detection rate is presented for each of the nine clusters (A to I). The right-hand side insets indicate the relative population size of each clusters. The best-correlating microseismic energy have been reported on top of clusters C and D, respectively identified from offshore the city of Nuugastiaq, and in the middle of the North Atlantic (see Supplementary Note 3 and Supplementary Figs. 2 and 3 for more details).

additional clustering procedure detected two clusters presented in Fig. 6a. These two clusters have different temporal cumulated detections and exhibit different population sizes. A zoom of the cumulated within-cluster detections is presented in Fig. 6b, and show a high similarity with clusters 3 and 4 previously obtained in Fig. 3 from the daylong seismogram. This result clearly proves

that the accelerating precursor is captured by our strategy even when the data is highly imbalanced. If the scattering network provides highly relevant features, clustering the seismic data with simple clustering algorithms can be a hard task that can be solved with hierarchical clustering, as illustrated in this study. This problem can also be better tackled by other clustering algorithms

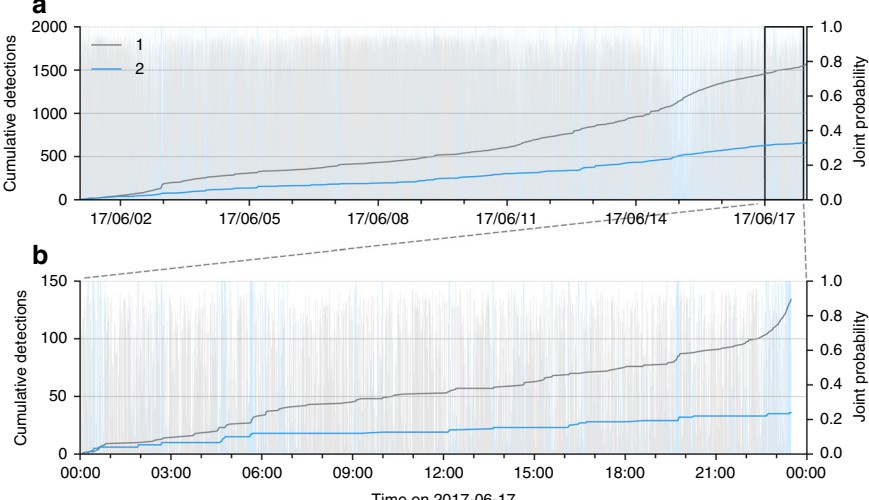

**Fig. 6 Hierarchical clustering of long-duration seismic data. a** Within-cluster cumulative detection overseen for second-order clustering of former clusters F to I presented in the Supplementary Fig. 1 from June 1, 2017 to June 18, 2017. **b** Zoom on June 17, 2017 from the detections presented in **a**. Similar to Fig. 3, the relative probability for each time window to belong to each cluster is represented with lighter bars.

such as spectral clustering[43], which has the additional ability to detect outliers. Clustering the outlier signals may then be an alternative to GMM in that case. Another possibility would be to use the local similarity search with hashing functions[15] in order to improve our detection database on large amount of the seismic data.

The structure of the scattering network shares some similarities with the FAST algorithm (for fingerprint and similarity search[15]) from a architectural point of view. FAST uses a suite of deterministic operations in order to extract waveforms features and feed it to a hashing system in order to perform a similarity search. The features are extracted from the calculation of the spectrogram, Haar wavelet transforms and thresholding operations. While being similar, the FAST algorithm involves a number of parameters that are not connected to the underlying physics. For instance, the thresholding operation has to be manually inspected[15], as well as the size of the analyzing window. In comparison, our architecture and weights are physically informed, and do not imply any signal windowing (only the resolution of the final result can be controlled). FAST is not a machine-learning strategy because no learning is involved; in contrast, we do learn the representation of the seismic data that best solves the task of clustering. While FAST needs a large amount of data to be run in an optimal way[15], our algorithm still works with a few number of samples.

This work shows that learning a representation of the seismic data in order to cluster seismic events in continuous waveforms is a challenging task that can be tackled with deep learnable scattering networks. The blind detection of the seismic precursors to the 2017 Landslide of Nuugaatsiaq with a deep learnable scattering network is a strong evidence that weak seismic events of complex shape can be detected with a minimum amount of prior knowledge. Discovering new classes of seismic signals in continuous data can, therefore, be better addressed with such strategy, and could lead to a better forecasting of the seismic activity in seismogenic areas.

## Methods

**Deep scattering network**. A complex wavelet $\psi \in \mathcal{L}$ is a filter localized in frequency with zero average, center frequency $\omega_0$, and bandwidth $\delta\omega$. We define the functional space $\mathcal{L}$ of any complex wavelet $\psi$ as

$$\mathcal{L} = \left\{ \psi \in L_c^2(\mathbb{C}), \int \psi(t)\mathrm{d}t = 0 \right\}, \qquad (1)$$

where $L_c^2(\mathbb{C})$ represents the space of square-integrable functions with compact time support $c$ on $\mathbb{C}$. At each layer, the mother wavelet $\psi_0 \in \mathcal{L}$ is used to derive a number of $JQ$ wavelets of the filterbank $\psi_j$ with dilating the mother wavelet by means of scaling factors $\lambda_j \in \mathbb{R}$ such as

$$\psi_j(t) = \lambda_j \psi_0(t\lambda_j), \quad \forall j = 0 \ldots JQ - 1. \qquad (2)$$

where the mother wavelet is centered at the highest possible frequency (Nyquist frequency). The scaling factor $\lambda_j = 2^{-j/Q}$ is defined as powers of two in order to divide the frequency axis in portions of octaves, depending on the desired number of wavelets per octave $Q$ and total number of octave $J$, which controls the frequency-axis limits and resolution at each layer. The scales are designed to cover the whole frequency axis, from the Nyquist angular frequency $\omega_0 = \pi$ down to a smallest frequency $\omega_{QJ-1} = \omega_0 \lambda_J$ defined by the user.

We define the first convolution layer of the scattering network (conv1 in Fig. 2) as the convolution of any signal $x(t) \in \mathbb{R}^C$ (where $C$ denotes the number of channels) with the set of $J^{(1)}Q^{(1)}$ wavelet filters $\psi_j^{(1)}(t) \in \mathcal{L}$ as

$$U_j^{(1)}(t) = \left| x * \psi_j^{(1)} \right|(t) \in \mathbb{R}^{C \times J^{(1)} \times Q^{(1)}}, \qquad (3)$$

where $*$ represents the convolution operation. The first layer of the scattering network defines a scalogram, a time-frequency representation of the signal $x(t)$ according to the shape of the moher wavelet $\psi_0^{(1)}$ widely used in the analysis of one-dimensional signals, including seismology.

The first-order scattering coefficients $S_j^{(1)}(t)$ are obtained after applying an average-pooling operation $\phi(t)$ over time to the first-order scalogram $U_j^{(1)}(t)$

$$S_j^{(1)}(t) = \left( U_j^{(1)} * \phi_1 \right)(t) = \left( \left| x * \psi_{j_1} \right| * \phi_1 \right)(t). \qquad (4)$$

The average-pooling operation is equivalent to a low-pass filtering followed by a downsampling operation[35]. It ensures the scattering coefficients to be locally stable with respect to time, providing a representation stable to local deformations and translations[21]. This property is essential in the analysis of complex signals such as seismic signals that can often be perturbed by scattering or present a complex source time function.

The small detailed information that has been removed by the pooling operation with Eq. (4) could be of importance to properly cluster different seismic signals. It is recovered by cascading the convolution, modulus, and pooling operations on higher-order convolutions performed on the first convolution layer (thus defining the high-order convolution layers shown in Fig. 2):

$$S_j^{(\ell)}(t) = U_j^{(\ell)}(t) * \phi_j^{(\ell)}(t), \qquad (5)$$

where $U^{(0)}(t) = x(t)$ is the (possibly multichannel) input signal (Fig. 2). The scattering coefficients are obtained at each layers from the successive convolution of the input signal with different filters banks $\psi^{(\ell)}(t)$. In addition, we apply an average-pooling operation to the output of the convolution-modulus operators in

order to downsample the successive convolutions without aliasing. This allow for observing larger and larger timescales in the structure of the input signal at reasonable computational cost.

We define the relevant features $\mathbf{S}(t)$ of the continuous seismic signal to be the concatenation of all-orders scattering coefficients obtained at each time $t$ as

$$\mathbf{S}(t) = \{S^{(\ell)}\}_{\ell=1\dots M} \in \mathbb{R}^F, \tag{6}$$

with $M$ standing for the depth of the scattering network, and $F = J^{(1)}Q^{(1)}(1 + \dots (1 + J^{(M)}Q^{(M)}))$ is the total number of scattering coefficients (or features). When dealing with multiple-channel data, we also concatenate the scattering coefficients obtained at all channels. The feature space therefore is a high-dimensional representation that encodes multiple time-scale properties of the signal over a time interval $[t, t + \delta t]$. The time resolution $\delta t$ of this representation then depends on the size of the pooling operations. The choice of the scattering network depth thus should be chosen so that the final resolution of analysis is larger than maximal duration of the analyzed signals.

Seismic signals can have several orders of different magnitude, even for signals lying in the same class. In order to make our analysis independent from the amplitude, we normalize the scattering coefficient by the amplitude of their "parent". The scattering coefficients of order $m$ are normalized by the amplitude of the coefficients $m - 1$ down to $m = 2$. For the first layer (which has no parent), the scattering coefficients are normalized by the coefficients of the absolute value of the signal[44].

**Adaptive Hermite cubic splines**. Instead of learning all the coefficients of the mother wavelet $\psi_0^{(\ell)}$ at each layer in the frequency domain, as one would do in a convolutional neural network, we restrict the learning to the amplitude and the derivative on a specific set of $\mathcal{K}$ knots $\{t_k \in c\}_{k=1\dots\mathcal{K}}$ laying in the compact temporal support $c$ (see Eq. (1)). The mother wavelet $\psi_0^{(\ell)}$ can then be approximated with Hermite cubic splines[23], a third-order polynomial defined on the interval defined by two consecutive knots $\tau_k = [t_k, t_{k+1}]$. The four equality constraints

$$\begin{cases} \psi_0^{(\ell)}(t_k) = \gamma_k \\ \psi_0^{(\ell)}(t_{k+1}) = \gamma_{k+1} \\ \dot{\psi}_0^{(\ell)}(t_k) = \theta_k \\ \dot{\psi}_0^{(\ell)}(t_{k+1}) = \theta_{k+1} \end{cases}, \tag{7}$$

uniquely determine the Hermite cubic spline solution piecewise on the consecutive time segments $\tau_k$, given by

$$\psi_{0,\Gamma,\Theta}^{(\ell)}(t) = \sum_{k=1}^{\mathcal{K}-1} \gamma_k f_1(x_k(t)) + \gamma_{k+1} f_2(x_k(t)) + \theta_k f_3(x_k(t)) + \theta_{k+1} f_4(x_k(t)) \mathbf{1}_{\tau_k}, \tag{8}$$

where $\Gamma = \{\gamma_k\}_{k=1\dots\mathcal{K}-1}$ and $\Theta = \{\theta_k\}_{k=1\dots\mathcal{K}-1}$, respectively, are the set of value and derivative of the wavelets on the knots, where $x(t) = \frac{t-t_k}{t_{k+1}-t_k}$ is the normalized time on the interval $\tau_k$, and where the Hermite cubic functions $f_i(t)$ are defined as

$$\begin{cases} f_1(t) = 2t^3 - 3t^2 + 1, \\ f_2(t) = -2t^3 + 3t^2, \\ f_3(t) = t^3 - 2t^2 + t, \\ f_4(t) = t^3 - 2t^2. \end{cases} \tag{9}$$

We finally ensure that the Hermite spline solution lays in the wavelets functional space $\mathcal{L}$ defined in Eq. (1) by additionally imposing

- the compactness of the support: $\gamma_1 = \theta_1 = \theta_{\mathcal{K}} = \gamma_{\mathcal{K}} = 0$,
- the null average: $\gamma_k = -\sum_{n \neq k} \gamma_n$,
- that the coefficients are bounded: $\max_t \gamma_t < \infty$.

The parameters $\gamma_k$ and $\theta_k$ solely control the shape of the mother wavelet, and are the only parameters that we learn in our strategy. Thanks to the above constraints, for any value of those parameters, the obtained wavelet is guaranteed to belong into the functional space of wavelets $\mathcal{L}$ defined in Eq. (1) with compact support. By simple approximation argument, Hermite cubic splines can approximate arbitrary functions with a quadratically decreasing error with respect to the increasing number of knots $\mathcal{K}$. Once the mother filter has been interpolated, the entire filterbank is derived according to Eq. (2).

**Clustering in a low-dimensional space**. We decompose the scattering coefficients $\mathbf{S}$ onto its two first-principle components by means of singular value decomposition $\mathbf{S} = \mathbf{UDV}^\dagger$, where $\mathbf{U} \in \mathbb{R}^{F \times F}$ and $\mathbf{V} \in \mathbb{R}^{T \times T}$ are, respectively, the feature- and time-dependant singular matrices gathering the singular vectors column-wise, $\mathbf{D}$ are the singular values, and where $T$ is the total number of time samples in the scattering representation. We define the latent space $\mathbf{L} \in \mathbb{R}^{2 \times T}$ as the projection of the scattering coefficients onto the first two feature-dependent singular vectors. Noting $\mathbf{U} = \{\mathbf{u}_i\}_{i \in [1\dots F]}$ and $\mathbf{V} = \{\mathbf{v}_j\}_{j \in [1\dots T]}$, where $\mathbf{u}_i$ and $\mathbf{v}_j$ are, respectively, the

singular vectors, the latent space is defined as

$$\mathbb{R}^{2 \times T} \ni \mathbf{L} = \sum_{i=1}^{2} \mathbf{Su}_i \tag{10}$$

To tackle clustering tasks, it is common to resort to centroidal-based clustering. In such strategy, the observations are compared with cluster prototypes and associated to the clusters with prototype the closest to the observation. The most-famous centroidal clustering algorithm is probably the $K$-means algorithm. Its extension, the Gaussian mixture model extends it by allowing nonuniform prior over the clustering (unbalanced in the clusters) and by allowing adapting the metric used to compare an observation to a prototype by means of a covariance matrix. To do so, Gaussian mixture model resorts to a generative modeling of the data. When using a Gaussian mixture model, the data are assumed to be generated according to a mixture of $K$-independent normal (Gaussian) processes $\mathcal{N}(\mu_k, \Sigma_k)$ as in

$$x \sim \prod_{k=1}^{K} \mathcal{N}(\mu_k, \Sigma_k) \mathbf{1}_{\{t=k\}}, \tag{11}$$

where $t$ is a Categorical variable governed by $t \sim \mathcal{C}at(\pi)$. As such, the parameters of the model are $\{\mu_k, \Sigma_k, k = 1\dots K\} \cup \{\pi\}$. The graphical model is given by $p(x, t) = p(x|t)p(t)$ and the parameters are learned by maximum likelihood with the expectation–maximization technique, where for each input $x$, the missing variable (unobserved) $t$ is inferred using expectation with respect to the posterior distribution as $E_{p(t|x)}(p(x|t)p(t))$. Once this latent variable estimation has been done, the parameters are optimized with their maximum-likelihood estimator. This two-step process is then repeated until convergence that is guaranteed[45].

**Learning the wavelets with gradient descent**. The clustering quality is measured in term of negative log-likelihood $\mathcal{T}$ with respect to the Gaussian mixture model formulation (here calculated with the expectation–minimization method). The negative log-likelihood is used to learn and adapt the Gaussian mixture model parameters (via their maximum-likelihood estimates) in order to fit the model to the data. We aim at adapting our learnable scattering filterbanks in accordance to the clustering task to increase the clustering quality. The negative log-likelihood will thus be used to adapt the filter-bank parameters.

This formulation alone contains a trivial optimum at which the filterbanks disregard any nonstationary event leading to a trivial single cluster and the absence of representation of any other event. This would be the simplest clustering task and would minimize the negative log-likelihood. As such, it is necessary to force the filterbanks to not just learn a representation more suited for Gaussian mixture model clustering but also not to disregard information from the input signal. This can be done naturally by enforcing the representation of each scattering to contain enough information to reconstruct the layer input signal. Thus, the parameters of the filters are learned to jointly minimize the negative log-likelihood and a loss of reconstruction.

**Reconstruction loss**. The reconstruction $\hat{x}(t)$ of any input signal $x(t)$ can be formally written in the single-layer case as

$$\hat{x}(t) = \sum_{i=1}^{JQ} \frac{1}{C(\lambda_i)} \sum_{t'} \psi_i(t - t') |(x * \psi_i)(t')|, \tag{12}$$

where $C(\lambda_i)$ is a renormalization constant at scale $\lambda_i$, and * stands for convolution. While some analytical constant can be derived from the analytical form of the wavelet filter, we instead propose a learnable coefficient obtained by incorporating a batch-normalization operator. The model thus considers $\hat{x} = (\text{BatchNorm} \circ \text{Deconv} \circ | \cdot | \circ \text{BatchNorm} \circ \text{Conv})(x)$. From this, the reconstruction loss is simply given by the expression

$$\mathcal{L}(x) = \| x - \hat{x} \|_2^2. \tag{13}$$

We use this reconstruction loss for each of the scattering layers.

**Stochastic gradient descent**. With all the losses defined above, we are able to leverage some flavor of gradient descent[39] in order to learn the filter parameters. Resorting to gradient descent is here required as analytical optimum is not available for the wavelet parameters, as we do not face a convex optimization problem. During training, we thus iterate over our data set by means of minibatches (a small collection of examples seen simultaneously) and compute the gradients of the loss function with respect to each of the wavelet parameters as

$$G(\theta) = \frac{1}{|\mathcal{B}|} \sum_{n \in \mathcal{B}} \left( \frac{\partial \mathcal{T}}{\partial \theta}(x_n) + \sum_{i=1}^{\ell} \frac{\partial \mathcal{L}^{(i)}}{\partial \theta}\left(x_n^{(i)}\right) \right), \tag{14}$$

with $\mathcal{B}$ being the collection of indices in the current batch, and $\theta$ being one of the wavelet parameters (the same is performed for all parameters of all wavelet layers). The $\ell$ superscript on the reconstruction loss represent the reconstruction loss for layer $\ell$. Then, the parameter is updated following

$$\theta^{t+1} = \theta^t - \alpha G(\theta), \tag{15}$$

with $\alpha$ the learning rate. Doing so in parallel for all the wavelet parameters

concludes the gradient descent update of the current batch at time $t$. This is repeated multiple time over different minibatches until convergence.

## Data availability

The facilities of IRIS Data Services, and specifically the IRIS Data Management Center, were used for access to waveforms and related metadata used in this study. IRIS Data Services are funded through the Seismological Facilities for the Advancement of Geoscience and EarthScope (SAGE) Project funded by the NSF under Cooperative Agreement EAR-1261681. The maps were made with the Cartopy Python library (v0.11.2. 22-Aug-2014. Met Office.). The topographic models were downloaded from the Global Multi-Resolution Topography databse at https://www.gmrt.org.

## Code availability

The codes used in the present study are freely available online at https://github.com/leonard-seydoux/scatnet.

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

## Acknowledgements

L.S., P.P., and M.C. acknowledge support from the European Research Council under the European Union Horizon 2020 research and innovation program (grant agreement no. 742335, F-IMAGE). M.C. and L.S. acknowledge the support of the Multidisciplinary Institute in Artificial Intelligence MIAI@Grenoble Alpes (Program "Investissements d'avenir" contract ANR-19-P3IA-0003, France). M.V.d.H. gratefully acknowledges support from the Simons Foundation under the MATH + X program and from DOE under grant DE-SC0020345. R.B. and R.G.B. were supported by NSF grants IIS-17-30574 and IIS-18-38177, AFOSR grant FA9550-18-1-0478, ONR grant N00014-18-12571, and a DOD Vannevar Bush Faculty Fellowship, ONR grant N00014-18-1-2047. L.S. thanks Romain Cosentino for very helpful discussions and comments.

## Author contributions

M.C. and M.V.d.H. initiated the study. P.P. proposed the case study. L.S. and R.B. implemented the codes and performed the training. L.S., M.C., and P.P. wrote the "Results" and "Discussion". R.B, L.S., M.V.d.H, and R.G.B. wrote the "Methods" section. All authors contributed to the "Abstract" and "Introduction".

## Competing interests

The authors declare no competing interests.
