## [Peer Review File · Nature Communications]

Reviewers' comments:

Reviewer #1 (Remarks to the Author):

The authors introduce a novel approach, using deep scattering networks with a cluster-based loss, together with Gaussian mixture model, to perform unsupervised clustering of seismic signals. As a proof-of-concept case study, the authors cluster signals in a small data set of continuous seismic data leading up to the 2017 Nuugaatsiaq landslide. The proposed approach is novel for the application in seismic signal processing and certain key properties of the method make it a promising alternative to many of the supervised deep learning methods proposed in the literature that require large labelled data sets. I believe this work is interesting, and has the potential to be a high impact contribution to the literature. However, the manuscript is not ready for publication without substantial revisions: as written the manuscript feels incomplete and does not address key points motivated in the introduction, specifically the scalability of the proposed method to large data sets.

The motivation for the work, as described in the abstract and introduction of the manuscript, emphasizes the need for analysis techniques that can be applied to seismic data sets that are 1) large and have 2) few/biased labels. Only the second of these two challenges is directly addressed: the proposed method handles unlabeled (or biased) data using an unsupervised learning approach. The conclusion reiterates that the proposed approach leads to "an unsupervised way of exploring large datasets." However, the manuscript does not explicitly address (through demonstration or discussion) how the proposed method tackles the challenge posed by "massive volumes of data" – a major omission.

The proof-of-concept case study applies the method to a very small data set, and the manuscript is missing any discussion of how the proposed method could scale to a larger data set in practice. For example, the training procedure for the scattering network requires clustering the data with a GMM – How does the training time scale with the size of the data set and are there any potential bottlenecks? Are there any computational tricks needed to scale up the approach from 24 hours of data to weeks, months, or years of data? For larger, more diverse data sets, a larger number of initial clusters would likely be necessary – How should this value be selected in practice? Will this increase the number of epochs required for convergence of the clustering loss? Will the inherent class-imbalance create challenges as the method is scaled to larger data sets and how might that be addressed?

Without addressing scalability, it is not clear what capability or advantage the proposed method has over existing methods (e.g. template matching, autocorrelation). For instance, the autocorrelation approach is cited as having poor scaling properties, but it is feasible to apply this method to a small data set of 24 hours.

Additional comments:

There is a disconnect between the title and body of the manuscript. The title suggests that the proposed method enabled a new scientific result, but the text does not discuss the scientific implications of the result (accelerated preparation process) from the analysis of the landslide case study.

The size of the data set used in the case study is not explicitly stated and can only be inferred through inspection of the figures (24 hours of continuous data and 2432 data points?). This information needs to be given explicitly in the text.

In the landslide case study, the results from the proposed method are compared to a correlation-based approach, but the details of how performance is evaluated is not clearly presented [Page

12]. This calls into questions the validity of the interpretation that the proposed method is more "discriminative" than a correlation-based approach.

- What is used as the ground truth for the performance evaluation? What, if anything, was done to verify that all of the signals in cluster 3 are real events? Were they visually inspected by a seismologist or verified by some other means?

- Some of the signals in cluster 3 are low SNR and not detected by template matching. What, if anything, was done to eliminate the possibility that the cluster contain both true events and background signals (especially since the clusters, Fig. 3, are not particularly compact)? These low SNR events are cited as evidence that the proposed method has "stronger discriminative power," but an alternative explanation is that these additional signals are not true events and that the proposed method has low precision. The change in time of the detection rate is not convincing evidence these are seismic events, as properties of background signals can also vary in time.

- Does the proposed method recover all signals identified by template matching?

The manuscript contains several overly general or unsupported claims:

- [Page 2] "unsupervised methods are more relevant for seismology" – unsupervised methods are certainly relevant, and perhaps more relevant for some tasks or data sets, but not all tasks.

- [Page 13] "without the need for any manual inspection" – certainly less manual inspection is required, but the clustering still requires manual interpretation (e.g. to determine which clusters contain signals of interest).

- The statement that the proposed method could lead to "informative/better forecasting" [Pages 1, 15] is offered without any supporting discussion.

Additional references:

- Literature review section [Page 3] should reference related work using deep learning for seismic signal detection, even if these studies have largely been using supervised methods.

- The manuscript should reference Mousavi et al. (2019) "Unsupervised clustering of seismic signals using deep convolutional autoencoders," IEEE Geoscience and Remote Sensing Letters, which also uses a deep neural network with a clustering loss to learn a representation for seismic signals [Page 4].

The manuscript contains some unclear or awkward phrasings or word choices, which at times makes it difficult to understand the meaning of the text. The manuscript may benefit from an additional read-through by a native English speaker. Some specific suggestions are listed under minor comments (see the attached document).

Reviewer #2 (Remarks to the Author):

In this paper the authors introduce a new algorithm to cluster seismic events based on a scattering transform. A scattering transform represents a signal by a low-dimensional vector of "features" computed by cascading wavelet transforms and modulus non-linearities. It linearises important transformations such as deformations and is therefore well adapted to a large range of classification tasks. The authors apply this transform to seismic signals and perform a clustering with a Gaussian mixture model.

The main originality of this work is that the authors learn the wavelets instead of using a predefined Gabor wavelet. This has never been done and they show that it considerably improve the classification results, which is quite remarkable. As a result, they obtain a convincing clustering results when applied on seismic data of the Nuugaatsiaq, Greenland landslide.

This paper thus presents a novel methodological approach to analyse seismic data which seems to provide very good results and I thus recommend its publication.

Reviewer #3 (Remarks to the Author):

In their manuscript, the authors apply a deep scattering convolutional neural network along with a Gaussian mixture model to detect earthquake precursors from the Nuugaatsiaq landslide. The algorithm appears to be successful, and detects precursors that have previously been identified with other algorithms. While it is nice that an unsupervised deep learning technique can achieve this type of detection, it is not clear that the algorithm is superior to other simpler, standard techniques (see below) nor is it clear that the algorithm is substantially different from certain other techniques (see below). Moreover, the implication that forecasting could be better done is unsubstantiated, since the algorithm requires a large number of similar events to exist before the clustering algorithm can identify the group. As such, it is unclear what the main new, interesting result is, and I suggest that the authors think more carefully about how to describe their results in the context of the existing literature and techniques.

Perhaps one of the biggest issues is that on p.5 the authors state that the precursory "signal is not directly visible in raw seismic records" and that this motivates the necessity for blind unsupervised strategies for detecting the landslide precursors. However, this statement is demonstrably false. Figure 2c of Poli (2017) (which is cited) clearly shows raw seismic records with obvious precursors. This figure suggests that a suitably tuned STA-LTA (short term average-long term average) standard earthquake detection algorithm (a very simple type of unsupervised algorithm) would have detected the precursors, and possibly with better fidelity than the authors' preferred algorithm. (Since the authors' algorithm has multiple tuning parameters, the tuning needed for a successful STA-LTA detection would not be a relative deficiency.) The need for a better algorithm for the Nuugaatsiaq landslide case study or cases like it is therefore unclear. There may be examples where the deep scattering network is needed and/or better than alternatives, but the case is not made clearly enough by the authors.

Another significant issue is that there appears to be significant similarity between the deep scattering network algorithm and the existing "FAST" detection routine of Yoon et al. (2015) (which is cited), which has already been applied in similar scenarios. While the algorithms have a number of differences, they have some of the same main qualitative features, including first using a frequency-time domain transform (spectrogram in FAST, wavelet in the present work), downsampling and thresholding of the results, and then a second wavelet transform. Although the FAST algorithm is not constructed as a neural network, the similarity between the algorithms deserves discussion, particularly if the authors are claiming their method is better. Ideally, the two algorithms would be compared side by side, though this is not necessary if the authors can convincingly describe why the scattering algorithm is sufficiently different.

In addition to the main points described above, the language throughout could be improved substantially. In a number of places, the implications are unclear due to incomplete description, awkward wording, or occasionally incorrect or misleading statements. As just a few examples, on p.2 "cannot be used ..." is inaccurate because it depends on the class of signals, "relevant for seismology" should more accurately be "relevant for discovery in seismology", on p.6 "by cascading" is (technically) unclear, "obtain a locally stable signal" is also unclear, on p.7 "often limited to a typical time scale" appears to be false at least for some common/routine algorithms, "analysis the filter" is unclear, "cross-validated onto audio" is unclear, on p.12 "In contrast, ..." is unclear, "striking" is excessive, and on p.15 "better forecasting" seems poorly motivated. Finally, Fig.4 could have been described better. For example, in the text there is mention that group 3 has a "brutal change" at 07:00 but I do not understand what the authors are pointing to, as I do not see any sudden change anywhere in Fig4B. This needs clarification.

Reviewer #1 (Remarks to the Author):

The authors introduce a novel approach, using deep scattering networks with a cluster-based loss, together with Gaussian mixture model, to perform unsupervised clustering of seismic signals. As a proof-of-concept case study, the authors cluster signals in a small data set of continuous seismic data leading up to the 2017 Nuugaatsiaq landslide. The proposed approach is novel for the application in seismic signal processing and certain key properties of the method make it a promising alternative to many of the supervised deep learning methods proposed in the literature that require large labelled data sets. I believe this work is interesting, and has the potential to be a high impact contribution to the literature. However, the manuscript is not ready for publication without substantial revisions: as written the manuscript feels incomplete and does not address key points motivated in the introduction, specifically the scalability of the proposed method to large data sets.

The motivation for the work, as described in the abstract and introduction of the manuscript, emphasizes the need for analysis techniques that can be applied to seismic data sets that are 1) large and have 2) few/biased labels. Only the second of these two challenges is directly addressed: the proposed method handles unlabeled (or biased) data using an unsupervised learning approach. The conclusion reiterates that the proposed approach leads to “an unsupervised way of exploring large datasets.” However, the manuscript does not explicitly address (through demonstration or discussion) how the proposed method tackles the challenge posed by “massive volumes of data” – a major omission.

This remark pinpoints an essential aspect of our strategy that we indeed omitted to discuss. We should also emphasize that besides being able to handle large datasets with a few or biased labels, we also aim at providing a tool that can be designed straightforwardly.

We now dedicate two paragraphs to the scalability of the method (pages 16 and 17) in order to inform the reader about how to deal with large datasets. Because we use batch processing, we emphasize that the method scales linearly with the growing amount of data. We also discuss how this type of analysis may be affected by the possibly growing amount of seismic classes that may be present in the wavefield, please find this a detailed answer in our reply to the next comment.

On another hand, our tool is not only capable of dealing with large datasets but also with small ones, where most of the classical deep-learning-based strategies need for a large amount of data to be appropriately trained.

The proof-of-concept case study applies the method to a very small data set, and the manuscript is missing any discussion of how the proposed method could scale to a larger data set in practice. For example, the training procedure for the scattering network requires clustering the data with a GMM – How does the training time scale with the size of the data set and are there any potential bottlenecks? Are there any computational tricks needed to scale up the approach from 24 hours of data to weeks, months, or years of data? For larger, more diverse data sets, a larger number of initial clusters would likely be necessary – How should this value be selected in practice? Will this increase the number of epochs required for convergence of the clustering loss? Will the inherent class-imbalance create challenges as the method is scaled to larger data sets and how might that be addressed? Without addressing scalability, it is not clear what capability or advantage the proposed method has over existing methods (e.g. template matching, autocorrelation). For instance, the autocorrelation approach is cited as having poor scaling properties, but it is feasible to apply this method to a small data set of 24 hours.

Computationally speaking, we ensured that it is possible to go from a 24-hour dataset to a more substantial amount of data (months or years) through batch processing (now clearly stated in the discussion, page 16).

We also emphasize that at longer time scales, more seismic classes may be present, or even the background noise fluctuations may dominate the clustering results (as emphasized in Yoon et al., 2015 for the missing detections, and in the result presented in Fig. A1). The problem relates to a problem of time scales and imbalance, where a substantial amount of data samples are related to the fluctuating background noise, and where rare data samples are related to actual events. In order to avoid misleading and overcomplete manuscripts, we propose to investigate this problem in future studies using hierarchical clustering rather than Gaussian Mixture Model. In the attached Fig. A1, we applied our clustering strategy to 23 consecutive days before the landslide recorded at NUUG. We observe that the network does not capture anymore the precursory signal, and that remaining clusters structures are most likely related to the background noise fluctuations. This illustrates the limits of GMM to capture rare events in largely unbalanced data, and will be in the scope of future studies to implement more sophisticated strategy and understand what we can find in this background noise clustering.

The upper boundary of the number of clusters must be defined based on the expert empirical intuitions. Relatively to our previous answer, we cannot aim at discovering at the same time small events localized in time and large-scale variations in the background seismic noise. With this in mind, the operator usually aims at finding a reasonably low number of clusters depending on the dataset on interest. Again, finding more clusters in the data is more of a hierarchical clustering strategy.

Figure A1 – large-scale clustering of the data continuously recorded by the NUUG seismic stations. The same parameters for designing the scattering network were used in this study. The color indicated the cumulated probabilities to belong to each clusters – the larger is the belonging to a given cluster, the larger is the corresponding color band for each sample. Here, six clusters are found by the strategy. The time of the landslide is indicated with a red vertical bar. Data gaps are due to tsunami waves and related power issues.

Additional comments

- There is a disconnect between the title and body of the manuscript. The title suggests that the proposed method enabled a new scientific result, but the text does not discuss the scientific implications of the result (accelerated preparation process) from the analysis of the landslide case study.

We agree with this comment. We changed the title in order to show that we focus on signal detection rather than on physical modeling.

- The size of the data set used in the case study is not explicitly stated and can only be inferred through inspection of the figures (24 hours of continuous data and 2432 data points?). This information needs to be given explicitly in the text.

Indeed, this information was missing. We now clearly indicate which portion of the data we analyze (page 5, second paragraph) and that the final temporal resolution of the scattering coefficients (or equivalently the latent space) is of about 35 seconds (leading to more than 2400 points).

- In the landslide case study, the results from the proposed method are compared to a correlation-based approach, but the details of how performance is evaluated is not clearly presented [Page 12]. This calls into questions the validity of the interpretation that the proposed method is more “discriminative” than a correlation-based approach. What is used as the ground truth for the performance evaluation? What, if anything, was done to verify that all of the signals in cluster 3 are real events? Were they visually inspected by a seismologist or verified by some other means?

According to these comments, we made several changes in the manuscript in several places. First, we do not speak about ground truth because the only reference we have is the template matching catalog obtained in Poli 2017. Template matching provides an accurate similarity measurement of a waveform because the full waveform is considered in the procedure; we consider features instead. Here we do not aim at outperforming the detections made with template matching. Instead, we believe that the structure of the temporal within-clusters cumulative detection curves are informative. We observe the presence of an accelerating even in our clusters, even with different parameter sets, as shown in the supplementary material, showing that our procedure successfully recovers the signature of the precursory signals, which is the main object of our study.

- Some of the signals in cluster 3 are low SNR and not detected by template matching. What, if anything, was done to eliminate the possibility that the cluster contain both true events and background signals (especially since the clusters, Fig. 3, are not particularly compact)? These low SNR events are cited as evidence that the proposed method has "stronger discriminative power," but an alternative explanation is that these additional signals are not true events and that the proposed method has low precision. The change in time of the detection rate is not convincing evidence these are seismic events, as properties of background signals can also vary in time.

The belonging of each event to all clusters is provided in a statistical way, and we can observe in Fig. 4B and E that this probability fluctuates for different events. The idea to show this probability was to have an idea of the detection quality of our approach for each sample. We therefore agree that some events have a low probability to belong to cluster 3 and change the text accordingly. In addition, we do not claim anymore that our technique has a "stronger discriminative power" (page 13).

- Does the proposed method recover all signals identified by template matching?

Almost all (97%). This is now stated in the manuscript (page 13).

- [Page 2] “unsupervised methods are more relevant for seismology” – unsupervised methods are certainly relevant, and perhaps more relevant for some tasks or data sets, but not all tasks.

Indeed, not all task are to be solved with unsupervised strategies in seismology; we reformulate the text to emphasize that *"Seismology can significantly benefit from the development of unsupervised approaches because the data is mostly unlabeled. Unsupervised tools are likely the best candidates to explore seismic data without the need for any explicit signal model and hence, discover new classes of seismic signals."* (page 2 and 3).

- [Page 13] “without the need for any manual inspection” – certainly less manual inspection is required, but the clustering still requires manual interpretation (e.g. to determine which clusters contain signals of interest).

This formulation was indeed misleading. We changed to state that *"This indicates that waveform templates can be recovered from our method without the need of any manual inspection of the seismic data prior to the clustering process, and tedious selection of waveform template in order to perform high-quality detection."* (page 15)

- The statement that the proposed method could lead to “informative/better forecasting” [Pages 1, 15] is offered without any supporting discussion.

We consider that finding precursory signals in continuous time series is fundamentally an strategy that would allow to "better forecast" major ruptures.

- Literature review section [Page 3] should reference related work using deep learning for seismic signal detection, even if these studies have largely been using supervised methods.
- The manuscript should reference Mousavi et al. (2019) "Unsupervised clustering of seismic signals using deep convolutional autoencoders," IEEE Geoscience and Remote Sensing Letters, which also uses a deep neural network with a clustering loss to learn a representation for seismic signals [Page 4].

This is now done.

The manuscript contains some unclear or awkward phrasings or word choices, which at times makes it difficult to understand the meaning of the text. The manuscript may benefit from an additional read-through by a native English speaker. Some specific suggestions are listed under minor comments (see the attached document).

We gratefully thank reviewer 1 for all useful comments and remarks.

Reviewer #2 (Remarks to the Author):

In this paper the authors introduce a new algorithm to cluster seismic events based on a scattering transform. A scattering transform represents a signal by a low-dimensional vector of "features" computed by cascading wavelet transforms and modulus non-linearities. It linearises important transformations such as deformations and is therefore well adapted to a large range of classification tasks. The authors apply this transform to seismic signals and perform a clustering with a Gaussian mixture model.

The main originality of this work is that the authors learn the wavelets instead of using a predefined Gabor wavelet. This has never been done and they show that it considerably improve the classification results, which is quite remarkable. As a result, they obtain a convincing clustering results when applied on seismic data of the Nuugaatsiaq, Greenland landslide.

This paper thus presents a novel methodological approach to analyse seismic data which seems to provide very good results and I thus recommend its publication.

Reviewer #3 (Remarks to the Author):

In their manuscript, the authors apply a deep scattering convolutional neural network along with a Gaussian mixture model to detect earthquake precursors from the Nuugaatsiaq landslide. The algorithm appears to be successful, and detects precursors that have previously been identified with other algorithms. While it is nice that an unsupervised deep learning technique can achieve this type of detection, it is not clear that the algorithm is superior to other simpler, standard techniques (see below) nor is it clear that the algorithm is substantially different from certain other techniques (see below). Moreover, the implication that forecasting could be better done is unsubstantiated, since the algorithm requires a large number of similar events to exist before the clustering algorithm can identify the group. As such, it is unclear what the main new, interesting result is, and I suggest that the authors think more carefully about how to describe their results in the context of the existing literature and techniques.

Perhaps one of the biggest issues is that on p.5 the authors state that the precursory “*signal is not directly visible in raw seismic records*” and that this motivates the necessity for blind unsupervised strategies for detecting the landslide precursors. However, this statement is demonstrably false. Figure 2c of Poli (2017) (which is cited) clearly shows raw seismic records with obvious precursors. This figure suggests that a suitably tuned STA-LTA (short term average-long term average) standard earthquake detection algorithm (a very simple type of unsupervised algorithm) would have detected the precursors, and possibly with better fidelity than the authors’ preferred algorithm. (Since the authors’ algorithm has multiple tuning parameters, the tuning needed for a successful STA-LTA detection would not be a relative deficiency.) The need for a better algorithm for the Nuugaatsiaq landslide case study or cases like it is therefore unclear. There may be examples where the deep scattering network is needed and/or better than alternatives, but the case is not made clearly enough by the authors.

First, the precursory signal is not visible in the raw seismograms, only if we consider the filtered seismogram, and only at very late times (see the attached figure A1). The energy of the precursory activity increases over time (Poli, 2017), such as the amplitude exceeds the background noise variance after 21:30 approximately. In Poli (2017), Fig. 3C, we observe the precursory events a few minutes before the mainshock in the frequency band 2 to 9 Hz; this is why we can observe some precursors therein. For the previous precursory events, only a full-waveform based detection could

Figure A1 – broadband (top) and filtered (bottom) seismogram from the NUUG seismic station. The seismogram is shown on a larger time period (from 15:00 to 00:00) where detections were found in the current manuscript and Poli (2017).

have worked, such as template matching (Poli, 2017). Second, the point of our paper is not only to detect signals but also to perform clustering analysis. Even if STA/LTA could be able to detect some of those signals, it would not make any difference between several types of seismicity (e.g., the waveforms in clusters 3 and 4 found by our strategy), which would affect the acceleration behavior seen in our result. Last, we can reveal the precursory activity without the need to filter the signal first and extract a template. This will be of particular interest when investigating other landslides or ruptures and see if a similar pattern can be found in the continuous seismic data, even if these potential signals have a low signal-to-noise ratio. Again, tuning up STA/LTA could be a tedious task, and our strategy is a tool that could overcome this problem.

In order to clarify in the text, we rephrased the paragraph (end of page 5, page 6 and 7), where we develop the motivations of our work in comparison with the previous study of Poli (2017).

Another significant issue is that there appears to be significant similarity between the deep scattering network algorithm and the existing “FAST” detection routine of Yoon et al. (2015) (which is cited), which has already been applied in similar scenarios. While the algorithms have a number of differences, they have some of the same main qualitative features, including first using a frequency-time domain transform (spectrogram in FAST, wavelet in the present work), downsampling and thresholding of the results, and then a second wavelet transform. Although the FAST algorithm is not constructed as a neural network, the similarity between the algorithms deserves discussion, particularly if the authors are claiming their method is better. Ideally, the two algorithms would be compared side by side, though this is not necessary if the authors can convincingly describe why the scattering algorithm is sufficiently different.

Here again, this comment raises several points that we can develop here. Please see our answer below, which led to substantial modifications in the text, in particular, the second paragraph on page 17.

We now expose that our algorithm shares some similarities in terms of architecture, but in contrast, FAST is not a learning algorithm, ours is. FAST is intended to overcome the computational limitations of the "autocorrelation" strategy that would be to consider every time segment of a time series as a potential template and seek for a match in the continuous waveform; thus really intense to compute. FAST provides an alternative to the autocorrelation with searching a similarity between waveforms with local hashing similarity. FAST computes waveform fingerprints (e.g., features) extracted with a deterministic suite of functions (spectrogram > Haar transform > thresholding > binarization). All these operations imply several sets of parameters that the authors have selected based on visual inspection (see Yoon et al., 2015), mostly because most of the parameters have no real physical meaning (thresholding value, number of hash tables, hashing table size, etc.). In our case, the parameters can be defined based on some physical intuition (time scales seen by each wavelet bank, frequency span, and resolution). Also, we claim that our strategy does not need any pre-processing in the data. In FAST, the authors have to bandpass-filter the data before the application of the algorithm.

Second, we do not aim at comparing the performance of FAST and our strategy, because they are not intended to be used the same way. For instance, it is claimed by the authors (Yoon et al. 2015) that at least one week of data is necessary to start taking benefit of the FAST algorithm, with a probable maximal capacity over one year of data. In our case, we claim that our algorithm is scalable but also works with a small amount of data such as in the case presented in the paper, a significant advance in comparison with FAST, and generally other deep nets that often require a lot of data to be run properly. On another hand, instead of comparing FAST with the deep scattering network, we now claim that a combination of the automatic feature extraction with the local hashing similarity search could be a great way to combine the straightforwardness of the deep scattering network with the power of similarity search with hashing functions.

In addition to the main points described above, the language throughout could be improved substantially. In a number of places, the implications are unclear due to incomplete description, awkward wording, or occasionally incorrect or misleading statements. As just a few examples, on p.2 “cannot be used ...” is inaccurate because it depends on the class of signals, “relevant for seismology” should more accurately be “relevant for discovery in seismology”, on p.6 “by cascading” is (technically) unclear, “obtain a locally stable signal” is also unclear, on p.7 “often limited to a typical time scale” appears to be false at least for some common/routine algorithms, “analysis the filter” is unclear, “cross-validated onto audio” is unclear, on p.12 “In contrast, ...” is unclear, “striking” is excessive, and on p.15 “better forecasting” seems poorly motivated. Finally, Fig.4 could have been described better. For example, in the text there is mention that group 3 has a “brutal change” at 07:00 but I do not understand what the authors are pointing to, as I do not see any sudden change anywhere in Fig4B. This needs clarification.

"cannot be used..." (page 2): we want to stress here that if some form of seismic signals escapes our detection systems so far, it may be due to the limited models of signal that we traditionally use to detect or qualify seismic events. As an example, we cite the discovery of non-volcanic tremors (first paragraph, page 2) that was not considered a seismic event 20 years ago. If other, different signals still escape our detection algorithms, we state that no supervised algorithm can solve the task of finding it because we do not provide any example of such signals. Unsupervised strategies may, in contrast, be a solution to solve it. We reformulate the statement to: *"Supervised algorithms rely on the quality of the predefined labels, often obtained via classical algorithms or even manually. They are made for learning to detect or classify specific classes of signals, and, therefore, cannot be used for discovering new classes of seismic signals."*

"relevant for seismology" (page 2): indeed, this statement was not appropriate here. We reformulate to *"Seismology can greatly benefit from the development of unsupervised strategies because the data is largely unlabeled and the new classes of seismic signals should be sought."*

"cascading" (pages 3, 6, 7): although it was the terminology used by the authors in Bruna and Mallat (2014), we understand that the word "cascade" has an unclear meaning here technically speaking because it is also used in the deep learning community for the Cascading Correlation Neural Networks (a class of self-organizing neural nets). We used another formulation instead.

"locally stable signal representation": we now use the terminology "locally invariant" for better readability.

"striking" and "brutal change": three different trends are identified to describe the temporal behavior of cluster 3 in the preceding paragraph; a first nearly-linear trend, and second trend with a low detection rate that turns into a third accelerating trend. We do agree that the wording is a bit excessive here, we propose to use the term "interesting" instead of "striking", and to remove the word "brutal".

Reviewers' comments:

Reviewer #1 (Remarks to the Author):

I am satisfied with the revisions to the manuscript. I recommend this article for publication.

Reviewer #3 (Remarks to the Author):

While the manuscript is improved, and the authors have added discussion on each point requested by the reviewers, still some of the main deficiencies pointed out by both Reviewer 1 and myself (Reviewer 3) remain weak. Most of this can be addressed without further analysis and simply toning down some of the main claims. The methods used by the authors are indeed of wide interest to the community, but the authors need to be much more careful about what they suggest is possible if they do not actually have the evidence in hand (i.e. is future work). Most importantly, whether the new algorithm is a robust detection method without significant false positives remains unsubstantiated and it therefore remains unclear how much of the precursory seismicity detected is real.

Title: As pointed out by Reviewer 1, the title was not representative of the work, and it still is not. The most obvious reading of the current title is that the present manuscript 'revealed' accelerating seismicity that was not known previously. Obviously, this is not what the authors mean, since previous authors have already identified the accelerating seismicity. I would suggest the authors change the title to something like "Improving seismicity detection before the 2017 Nuugaatsiaq landslide with unsupervised deep learning"

p.5 "This signal is not directly visible in raw seismic records... The structure of such signal makes it hard to detect via traditional detection algorithms" This statement is still extremely misleading. The vast majority of seismicity that is routinely detected with standard seismological techniques is also not directly visible in raw seismic records. It is only after filtering that many of these signals appear visibly and for which a STA/LTA algorithm would be successful for, and the precursory seismicity being discussed is no exception. As mentioned in my original review, this makes the statement extremely misleading, even if the first sentence is not strictly incorrect. This discussion needs to be made fairer to existing 'blind, unsupervised' algorithms like standard STA/LTA algorithms, which the authors do not characterize accurately.

p.12 "The detection rate is sparse in time, indicating that cluster 4 is mostly related to a random background seismicity" This statement requires further justification and logic. The fact that it is sparse does not implicate seismicity. Unfortunately, seismometer response is affected by numerous processes other than earthquakes that may be sparse in time, including various animal behavior, instrument glitches, etc. Although this is not so important, some of this lack of justification also applies to cluster 3 which is more pertinent to the main points of the manuscript.

p.13 "accelerating seismicity is finally observed ... suggests that the repeating event may still exist earlier in the data" The interpretation of cluster 3 as precursory seismicity is still not properly justified, particularly for the time period before 15:00. Despite the added discussion, it is not clear that the majority of events detected are indeed related to anything precursory to the landslide. While the author response to reviewers file acknowledges this possibility, the manuscript text still strongly suggests most of the events in cluster 3 are real. What would be a good test of the authors' claim is if the same analysis were performed on a much longer time series (e.g. 2 weeks or 1 month), or even on one day of data far in advance of the landslide (e.g. 4-5 months earlier). If on this much earlier dataset, the authors find essentially no detections, while finding detections in the one day prior to the event, that would be much more convincing that the authors are not detecting other environmental processes unrelated to the landslide. Essentially, the present manuscript fails to show a controlled hypothesis, with a clear negative control for which no

seismicity is observed where it is not expected. Without such a control, I remain unconvinced that the algorithm is detecting any real precursory events that previous techniques did not detect or that the method can generally be used in any robust way for detection. If the authors want to temper their claims and restrict their claim to only being that their method recovers most (97%) of the events found by previous authors (and not about detecting more seismicity) plus other events that may not be real, then the title should be further restricted as well, perhaps to "An alternative method of seismicity detection before the 2017 Nuugaatsiaq landslide with unsupervised deep learning", so that the reader does not get the false impression that the authors believe it is better than other algorithms.

June 2, 2020

Object: Rebuttal letter for the manuscript NCOMMS-19-25635A

Title: Accelerating seismicity before the 2017 Nuugaatsiaq landslide revealed with unsupervised deep learning

Dear Reviewers,

Please find attached a new version of the manuscript NCOMMS-19-25635A entitled "Accelerating seismicity before the 2017 Nuugaatsiaq landslide revealed with unsupervised deep learning". In this new version we intended to address the main points raised by Reviewer 3 with adding an additional experiment over larger time scales in order to illustrate the scalability of our approach. In addition to retrieving the precursory seismic activity to the Nuugaastiaq landslide, we also show that the method is powerful to cluster long-duration seismic noises, in particular we show a correlation between the identified clusters and the microseismic activity related to oceanic storm systems.

Please find below a pointwise answer to Reviewer 3.

Reviewer 3 (Remarks to the Author):

While the manuscript is improved, and the authors have added discussion on each point requested by the reviewers, still some of the main deficiencies pointed out by both Reviewer 1 and myself (Reviewer 3) remain weak. Most of this can be addressed without further analysis and simply toning down some of the main claims. The methods used by the authors are indeed of wide interest to the community, but the authors need to be much more careful about what they suggest is possible if they do not actually have the evidence in hand (i.e. is future work). Most importantly, whether the new algorithm is a robust detection method without significant false positives remains unsubstantiated and it therefore remains unclear how much of the precursory seismicity detected is real.

The proposed method is presented as an exploratory tool to discover and efficiently identify patterns in continuous seismic seismograms. We now clearly state that the proposed approach is not simply be a substitute for existing detection techniques. The main idea of this manuscript is that the accelerating behaviour of the seismic precursor is correctly discovered by the unsupervised analysis, and suggest that it can be applied to other datasets in a blind and exploratory way. Once a particular behavior is identified, we clearly state that some further analysis can be carried out on the identified clusters, with full-waveform template matching for instance.

Title: As pointed out by Reviewer 1, the title was not representative of the work, and it still is not. The most obvious reading of the current title is that the present manuscript 'revealed' accelerating seismicity that was not known previously. Obviously, this is not what the authors mean, since previous authors have already identified the accelerating seismicity. I would suggest the authors change the title to something like "Improving seismicity detection before the 2017 Nuugaatsiaq landslide with unsupervised deep learning"

We do not fully agree on this title since our algorithm does not improve detection : the clustering results show that the precursory signal is detected and exhibit the accelerating shape in one of the obtained clusters. In this paper, we aim at showing that particular families of seismic waveforms can automatically be identified without any prior knowledge, and to illustrate that the relevant representations can be learned from the deep scattering network. This

study therefore allows to reveal the precursory signal, and to separate them from other types of seismicity (background noise, microseismic activity, random and sharp signals) but not to accurately detect each of the precursory events at once. As stated in multiple paragraphs of the introduction as well as in the Results sections, we do not aim at outperforming the already existing detection strategies, but to provide an exploratory tool for automatic identification of seismic signal types.

p.5 “This signal is not directly visible in raw seismic records... The structure of such signal makes it hard to detect via traditional detection algorithms” This statement is still extremely misleading. The vast majority of seismicity that is routinely detected with standard seismological techniques is also not directly visible in raw seismic records. It is only after filtering that many of these signals appear visibly and for which a STA/LTA algorithm would be successful for, and the precursory seismicity being discussed is no exception. As mentioned in my original review, this makes the statement extremely misleading, even if the first sentence is not strictly incorrect. This discussion needs to be made fairer to existing ‘blind, unsupervised’ algorithms like standard STA/LTA algorithms, which the authors do not characterize accurately.

As stated in lines 127 to 130 (p.5), our goal is not only to enhance the detection capacity, but to cluster seismic data. The discussion that we engaged in this paragraph does not only point out the requirement in human intervention for detecting such a signal with STA/LTA, but also the limitation to a detection power without any clustering capabilities. Again, triggering detectors such as STA/LTA strongly differ from our approach because of the lack of similarity measurement between the waveform. Besides, previous studies that made it possible to detect this signal had to use full-waveform characteristics in order to do so (template matching, Poli 2017), while STA/LTA was incapable to generate the same results. In our case, we do provide a detection algorithm, which in addition inform about the waveform similarity. We clarify this in lines 143 - 150.

p.12 “The detection rate is sparse in time, indicating that cluster 4 is mostly related to a random background seismicity” This statement requires further justification and logic. The fact that it is sparse does not implicate seismicity. Unfortunately, seismometer response is affected by numerous processes other than earthquakes that may be sparse in time, including various animal behavior, instrument glitches, etc. Although this is not so important, some of this lack of justification also applies to cluster 3 which is more pertinent to the main points of the manuscript.

Correct. We added a sentence to clarify that this may be related to background seismicity or other local signals recorded by the station that are not relevant to the present case study.

p.13 “accelerating seismicity is finally observed ... suggests that the repeating event may still exist earlier in the data” The interpretation of cluster 3 as precursory seismicity is still not properly justified, particularly for the time period before 15:00. Despite the added discussion, it is not clear that the majority of events detected are indeed related to anything precursory to the landslide. While the author response to reviewers file acknowledges this possibility, the manuscript text still strongly suggests most of the events in cluster 3 are real. What would be a good test of the authors’ claim is if the same analysis were performed on a much longer time series (e.g. 2 weeks or 1 month), or even on one day of data far in advance of the landslide (e.g. 4-5 months earlier). If on this much earlier dataset, the authors find essentially no detections, while finding detections in the one day prior to the event, that would be much more convincing that the authors are not detecting other environmental processes unrelated to the landslide. Essentially, the present manuscript

fails to show a controlled hypothesis, with a clear negative control for which no seismicity is observed where it is not expected. Without such a control, I remain unconvinced that the algorithm is detecting any real precursory events that previous techniques did not detect or that the method can generally be used in any robust way for detection. If the authors want to temper their claims and restrict their claim to only being that their method recovers most (97%) of the events found by previous authors (and not about detecting more seismicity) plus other events that may not be real, then the title should be further restricted as well, perhaps to “An alternative method of seismicity detection before the 2017 Nuugaatsiaq landslide with unsupervised deep learning”, so that the reader does not get the false impression that the authors believe it is better than other algorithms.

Indeed, we needed to clarify this point. We now present in the manuscript an additional experiment performed over a larger time window of signal before the landslide (17 days). Over this large period, the clustering algorithm still converges and exhibit highly imbalanced clusters (93% of noise versus 7% of signal). Again, for the sake of clarity, we never claim that we are able to isolate the precursory seismic signal in a dedicated cluster, but that all the seismic precursors are gathered in a single cluster. As we claim in the introduction, we do not aim at outperforming full-waveform similarity testing such as template-matching, and provide a tool that allow for clustering seismic waveforms based on wide-sense similarity, extracted from a straightforwardly designed deep neural network. The main point of the manuscript is to show that the accelerating behavior of the precursory activity is visible from the clustering procedure, that we illustrated with two different experiments in the new version. We therefore hope that the additional statements together with the new additional experiments presented in the main manuscript and in the supplementary materials have addressed Reviewers' 3 concerns.

Sincerely,

Anonymous authors

REVIEWERS' COMMENTS:

Reviewer #3 (Remarks to the Author):

There are still a number of points where the wording is misleading. Please see below.

The title is still inappropriately misleading and it is alarming that the authors do not appear to understand that the title strongly suggests that they are the ones to have first "revealed" the accelerating seismicity (which is factually untrue, which the authors acknowledge). Although it is a small change, for the sake of accuracy, I would insist that the title be changed, for example, to "Accelerating seismicity before the 2017 Nuugaatsiaq landslide can be revealed with unsupervised deep learning"

p.5 comment: I still disagree with the authors' contention that STA/LTA categorically has difficulty detecting the landslide seismicity. It is unclear whether an appropriately tuned STA/LTA algorithm would have been successful, primarily because such an approach was never attempted (in the literature or the present manuscript). I agree completely with the authors' response that STA/LTA detectors fall into a different category of routine. However, the authors should not state that they know that the seismicity would have been difficult to detect with an STA/LTA routine unless they have either a specific study which tried that and failed that they can cite or unless they tried it themselves. Although I have not attempted this either, my suspicion is that an appropriately tuned STA/LTA algorithm would also detect the seismicity in this case without any problem, which is why I think the current text is still misleading.

p.13 comment: "This suggests that the repeating event may still exist earlier in the data" This suggestion is still not substantiated. Given the new longer-term results shown in Fig6, it seems much more likely that the nearly linearly increasing number of detected events (category "1") in the 2-week time period prior to the clear acceleration are simply from random noise that occasionally gets categorized as "1" instead of "2" in Fig6. To be clear, I agree with the reviewers that the accelerating period is captured in category 1 (and not category 2), but not that the events prior to the accelerating time in category 1 are likely to be significantly different from those (e.g.) of category 2 and thus that they can be used to suggest that the repeating seismicity is possibly occurring for 2 weeks. The similarity of the category 1 and 2 curves for all but the last 2 hours in Fig.6 suggests there is some overlap in the types of events.